# Acute DOB and PMA Administration Impairs Motor and Sensorimotor Responses in Mice and Causes Hallucinogenic Effects in Adult Zebrafish

**DOI:** 10.3390/brainsci10090586

**Published:** 2020-08-24

**Authors:** Micaela Tirri, Luisa Ponzoni, Sabrine Bilel, Raffaella Arfè, Daniela Braida, Mariaelvina Sala, Matteo Marti

**Affiliations:** 1Department of Morphology, Surgery and Experimental Medicine, Section of Legal Medicine and LTTA Centre, University of Ferrara, 44121 Ferrara, Italy; micaela.tirri@unife.it (M.T.); sabrine.bilel@unife.it (S.B.); rfarfl@unife.it (R.A.); 2Department of Medical Biotechnology and Translational Medicine, Università degli Studi di Milano, 20133 Milan, Italy; luisa.ponzoni@guest.unimi.it (L.P.); daniela.braida@guest.unimi.it (D.B.); 3Institute of Public Health, Section of Legal Medicine, Catholic University of Rome, 00168 Rome, Italy; 4CNR, Neuroscience Institute, 20162 Milan, Italy; mariaelvina.sala@unimi.it; 5Collaborative Center for the Italian National Early Warning System, Department of Anti-Drug Policies, Presidency of the Council of Ministers, 00184 Rome, Italy

**Keywords:** DOB, hallucinogens, mice, PMA, prepulse inhibition, sensorimotor responses, zebrafish

## Abstract

The drastic increase in hallucinogenic compounds in illicit drug markets of new psychoactive substances (NPS) is a worldwide threat. Among these, 2, 5-dimetoxy-4-bromo-amphetamine (DOB) and paramethoxyamphetamine (PMA; marketed as “ecstasy”) are frequently purchased on the dark web and consumed for recreational purposes during rave/dance parties. In fact, these two substances seem to induce the same effects as MDMA, which could be due to their structural similarities. According to users, DOB and PMA share the same euphoric effects: increasing of the mental state, increasing sociability and empathy. Users also experienced loss of memory, temporal distortion, and paranoia following the repetition of the same thought. The aim of this study was to investigate the effect of the acute systemic administration of DOB and PMA (0.01–30 mg/kg; i.p.) on motor, sensorimotor (visual, acoustic, and tactile), and startle/PPI responses in CD-1 male mice. Moreover, the pro-psychedelic effect of DOB (0.075–2 mg/kg) and PMA (0.0005–0.5 mg/kg) was investigated by using zebrafish as a model. DOB and PMA administration affected spontaneous locomotion and impaired behaviors and startle/PPI responses in mice. In addition, the two compounds promoted hallucinatory states in zebrafish by reducing the hallucinatory score and swimming activity in hallucinogen-like states.

## 1. Introduction

In recent years, novel psychoactive substances (NPS) have emerged in response to legislative control and market trends [1]. NPS, also known as designer drugs, bath salts, or legal highs, are analogous or derivatives of controlled substances. Since 2012, the number of NPS seized in Europe has rapidly increased [2]. The large demand for stimulants and hallucinogens such as amphetamine, methamphetamine, 3, 4-metyhlendioxy methamphetamine (MDMA/Ecstasy), and LSD increased their production and trafficking in many EU Member States. The local synthesis and distribution of these substances limited their prohibition.

As a central European product, MDMA/ecstasy gained popularity among adults and also teenagers. In 2018, the number of European MDMA/ecstasy users (aged 15–64) reached approximately 13.7 million. MDMA use is more common among young people. In fact, it is estimated that 2% of young adults (aged 15–34) have used ecstasy in 2019 [3]. In addition, numerous NPS with stimulant and entactogenic properties have been designed to substitute MDMA and/or LSD in the illicit drug market. Among them are dimethoxylated-amphetamines such as 4-bromo-2, 5-dimethoxyamphetamine (DOB; Figure 1), which is a synthetic analogous of mescaline belonging to the class of phenethylamines. Recently, DOB gained more popularity as a drug of abuse in Italy [4]. This drug is usually sold online in the form of tablets and blotters under a variety of names such as “Golden” or “LSD-25”. Yet, DOB belongs to the D-series of substituted amphetamines with hallucinogen-like effects similar to that of LSD, high in vivo potency, and a long duration of action. However, their pharmacological effects are not fully established [5].

It has been reported that synthetic analogous of mescaline, including DOB, are more potent to induce hallucinations than those of many naturally occurring hallucinogens [6]. DOB interacts with serotonergic receptors, with the highest affinity for 5-HT2A receptors [7,8]. Studies in vivo revealed that DOB is 10 times more potent than 4-bromo-2, 5-dimethoxyphenethylamine (2C-B) in humans [9] and double/triple to the phenethylamine analogous (2C-B) in mice [10]. In addition, the behavioral effects of DOB seem to be 5-HT2A-dependent [11,12,13].

PMA (Figure 1) has been also designed to mimic the effects of MDMA. In fact, this drug is illegally sold in the form of powder or tablets and is labeled as “amphetamine” or “ecstasy”. The ring substitution enhances the serotonergic potency, hallucinogenic, and MDMA-like properties of methamphetamine/amphetamine drugs [14]. Incidental deaths have been reported after PMA and its derivative PMMA having a high toxicity, but systemic data are not available [15,16,17]. In particular, PMA appears to be more toxic than MDMA at low doses [18] and often, the effect is delayed by ingestion [19]. In vitro studies confirm that PMA potency is 100 times more potent than MDMA [20] and this is due to PMA’s ability to release and inhibit the reuptake of 5-HT [21].

Effects and dosage of DOB and PMA have been reported by psychonauts [22]. Both are identified as “pattern recognition enhancement” and considered as psychedelic substances with strong cognitive effects, even at low doses, as anxiety, agitation, mild paranoia, increased musical perception, temporal distortion, and double vision, are some of the most reported effects by users [23]. However, there are no currently available preclinical data on the major psychopharmacological effects of DOB and PMA.

On this basis, the purpose of this study is to evaluate in vivo the acute sensorimotor alteration and hallucinogenic properties, typical of phenethylamine and amphetamine-like compounds in the classical murine and the zebrafish models.

Recent evidence indicates that zebrafish is a sensitive and promising model to screen addicting drugs, showing high physiological and genetic homology to humans [24]. Moreover, zebrafish appears to be a sensitive and promising model for the investigation of hallucinogen-evoked states. In this regard, many hallucinogens (mescaline LSD, ketamine, PCP, MK-801, atropine, scopolamine, salvinorin A, and ibogaine) have been already reported in adult zebrafish [25].

Therefore, the effect of DOB and PMA was investigated on motor and sensorimotor (visual, acoustic, and tactile) responses and the ability to impair sensory gating reflexes using the prepulse inhibition (PPI) test in mice. Moreover, hallucinations were evaluated in terms of trance-like behavior, using a score scale according to Braida et al. [26] on adult zebrafish to confirm the hallucinogenic states induced by the two compounds.

## 2. Materials and Methods

### 2.1. Animals

Male ICR (CD-1^®^) mice, 3–4 months old, weighing 25–30 gr (ENVIGO Harlan Italy, Italy; bred inside the Laboratory for Preclinical Research (LARP) of University of Ferrara, Italy), were group-housed (5 mice per cage; floor area/animal: 80 cm^2^; minimum enclosure height: 12 cm) on a 12:12-h light–dark cycle (light on at 6:30 a.m.), temperature of 20–22 °C, humidity of 45–55% and were provided ad libitum access to food (Diet 4RF25 GLP; Mucedola, Settimo Milanese, Milan, Italy) and water. The experimental protocols were in accordance with the European Community Council Directive of September 2010 (2010/63/EU). A revision of the Directive 86/609/EEC was approved by the Ethics Committee of the University of Ferrara and by the Italian Ministry of Health (auth. num. 335/2016-PR). Moreover, adequate measures were taken to reduce the number of animals used and their pain and discomfort according to ARRIVE guidelines.

Adult wild-type short fin zebrafish (*Danio rerio*) (0.5–1 g, 6–12 months of age) of heterogeneous genetic background were obtained by a local aquarium supply store (Aquarium Center, Milan, Italy). All fish were housed in groups of 30 in a 96 L home tank maintained under standard conditions at 28 (±2) °C, 14–10 h day/night cycle (lights on at 7:00 a.m.) for at least 1 month after their arrival. Tank water, containing sea salts (Instant Ocean, Aquarium System, Sarrebourg, France) at a concentration of 0.6 g/10 L, was obtained by a reverse osmosis filter system. Water quality was maintained at optimal levels and checked daily for pH (zebrafish are maintained with a pH ranging from 6.6 to 8.2) and every 3 days for nitrates (<0.02 ppm) and nitrites (0.05–0.15 mg/L). Fish were fed twice a day with commercial flakes (tropical fish food, Consorzio G5, Italy) supplemented with live brine shrimp. In all of the experiments, the sex ratio was 50–50%. Males and females were identified according to Streisinger (2000). Behavioral testing took place during the light phase between 9.00 and 14.00 h. During the experiments, the observer, who was blind to the treatment allocation, sat 2 m away from the tank. All the fish were drug naïve, and each fish was used only once. Ten fish per group were used. A total of 150 zebrafish were used. All experiments were conducted in strict accordance with the European Community Council Directive No. 86/609/EEC and the subsequent Italian Law on the Protection of animals used for experimental and other scientific reasons. All efforts were made to minimize the number of animals used and their discomfort. All the behavioral experiments followed the ARRIVE guidelines.

### 2.2. Drug Preparation and Dose Selection

DOB and PMA were purchased from LGC Standards (LGC Standards, Milan, Italy).

Mouse. Drugs were initially dissolved in absolute ethanol (final concentration was 2%) and Tween 80 (2%) and brought to the final volume with saline (0.9% NaCl). The solution made of ethanol, Tween 80, and saline was also used as the vehicle [27,28]. Drugs were administered by intraperitoneal injection at a volume of 4 µL/g. In this trial, as already described in previous safety pharmacological studies on NPS [27,28,29], a wide range of doses was used for DOB (0.01–30 mg/kg) and PMA (0.01–30 mg/kg) for the purpose of better evaluating the behavioral effects of the two compounds on sensorimotor responses, prepulse inhibition studies, and spontaneous locomotion.

Zebrafish. DOB (0.075–2 mg/kg) and PMA (0.0005–0.5 mg/kg) (Sigma-Aldrich, St. Louis, MO, USA) were dissolved in sterile saline and administered immediately before each test. The vehicle group received saline (2 μL/g). The doses of the drugs were calculated as salt. All drugs were prepared fresh daily. As previously described [30], the dose range of DOB and PMA was chosen to better evaluate the behavioral effects of the two compounds on hallucinatory and swimming activity.

### 2.3. Behavioral Tests

For the overall study, 292 mice were used. In sensorimotor (visual object, visual placing, acoustic, and overall tactile responses) tests for MPA and PMA experiments, for each treatment (vehicle or 5 different DOB and PMA doses, 0.01, 0.1, 1, 10, and 30 mg/kg), 8 mice were used (total mice used: 96). In the analysis of spontaneous locomotion (open field test) for MPA and PMA experiments, for each treatment (vehicle or 5 different DOB and PMA doses, 0.01, 0.1, 1, 10, and 30 mg/kg), 8 mice were used (total mice used: 96), while for DOB and PMA experiments in the PPI test, for each treatment (vehicle or 4 different DOB and PMA doses, 0.01, 0.1, 1, and 10 mg/kg) 10 mice were used (total mice used: 100). The number of animals used was defined according to the 3R rules. In particular, the determination of the number of animals to be used (sample size) and the calculation of adequate power in the statistical analysis of the data (power analysis) were determined using the simulation software G * Power 3.1.9.2.

The effect of DOB and PMA on sensorimotor responses was investigated using a battery of behavioral tests widely used in studies of “safety-pharmacology” that were routinely adopted in our laboratory for the preclinical characterization of new molecules in rodents [31,32,33,34,35]. To reduce the number of animals used, mice were evaluated in functional observational tests carried out in a consecutive manner according to the following time scheme: observation of visual object responses (frontal and lateral view), acoustic response, tactile response (vibrissae, corneal, and pinnae reflexes), and visual placing response. Sensorimotor tests were measured at 10, 30, 60, 120, 180, 240, and 300 min after injections for the evaluation of the visual, acoustic, and tactile responses and at 15, 45, 75, 135, 195, 255, and 315 min after injections for the evaluation of visual placing response. Behavioral tests were conducted in a thermostated (temperature: 20–22 °C, humidity: 45–55%) and light (150 lux) controlled room with a background noise of 40 ± 4 dB (inside the LARP of University of Ferrara, Italy).

#### 2.3.1. Evaluation of the Visual Response

Visual response was verified by two behavioral tests, which evaluated the ability of the animal to capture visual information when this was either stationary (the visual object response) or moving (the visual placing response). For technical details of the methods used, see [32,33].

#### 2.3.2. Evaluation of Acoustic Response

Acoustic response measures the reflex of the mouse in response to an acoustic stimulus (four acoustic stimuli of different intensity and frequency were tested) produced behind the animal. For technical details of the methods used, see [32,33].

#### 2.3.3. Evaluation of Tactile Response

Tactile response in the mouse was verified through vibrissae, corneal, and pinnae reflexes. Data are expressed as the sum of the three abovementioned parameters. For technical details of the methods used, see [32,33].

#### 2.3.4. Spontaneous Locomotor Activity

Spontaneous locomotor activity was measured using the ANY-maze video tracking system (Ugo Basile, application version 4.99 g Beta). In the open field test, DOB and PMA were given intraperitoneally (i.p.) in mice, and each animal was singularly located in the open field box. For technical details of the methods used, see [32].

#### 2.3.5. Prepulse Inhibition (PPI) Test

Mice were tested for acoustic startle reactivity in startle chambers (Ugo Basile apparatus, Milan, Italy) consisting of a sound-attenuated, lighted, and ventilated enclosure holding a transparent non-restrictive Perspex^®^ cage (90 × 45 × 50 mm). All acoustic stimuli were produced through a loudspeaker mounted laterally to the holder. A load cell was then able to detect peak and amplitudes of the startle response. At the onset of the startling stimulus, 300-ms readings were recorded, and the wave amplitude evoked by the movement of the animal startle response was measured. For technical details of the methods used, see [28,33].

The test session contained 40 trials composed of pulse-alone (120 dB) and prepulse (120 dB) + pulse trials (of 68, 75, and 85 dB) presented in a pseudo-randomized order. DOB and PMA were administered intraperitoneally, and the first PPI test occurred at 15 min from drugs administration. The amount of PPI was expressed as the percentage decrease in the amplitude of the startle reactivity caused by the presentation of the prepulse (% PPI).

#### 2.3.6. Behavioral Analysis in Zebrafish

##### Intramuscular Injection

Zebrafish body weight was measured as previously described [26]. Fish were removed from their tank and placed in a tank containing water, located on a digital balance. Fish weight was calculated as the weight of the tank plus the fish minus the weight of the tank before the fish was added. Three measurements were recorded. Intramuscular (i.m.) injection was made in the caudal musculature along the posterior axis. Each fish received the injection in the area below the caudal fin on the left side. A Hamilton syringe (Hamilton Bonaduz AG, Bonaduz, Switzerland) delivered a volume, depending on the fish’s weight (2 μL/g). The fish were individually immobilized through the net with two fingers of the left hand. Then, the needle was positioned at a 45° angle in relation to the back of the fish with the needle pointing towards the head. The needle was inserted into the muscle just beyond the bevel of the needle. Total time during which each fish was out of water was approximately 10 s. After injection, each fish was immediately placed in the tank water.

##### Hallucinatory Behavior and Swimming Activity 

After injection, each subject was placed in an observation container (10 × 10 cm) with home tank water filled at a level of 12 cm. Immediately after injection, each fish was observed over a 30-min period every 5 min, for 30 s, for a total of six observation bins. A score scale was used to evaluate hallucinatory behavior as previously described [26]: 0—“trance-like” effect, horizontal motionless position on bottom tank maintained for 2–3 min at a time at the peak of the narcosis and broken by a brief change of position by means of a slight stimulus; 2—slowed swimming, normal body position; 4—normality state; 6—accelerated swimming, normal body position; 8—frenetic swimming, with the body suspended in the vertical or some angled from the vertical; 10—frenetic circling behavior.

In order to better quantify swimming activity, the number of crossed lines, made by each fish, was measured in the same tank used for the evaluation of hallucinatory behavior. This tank was previously, divided into ten equal-sized 2 × 10 cm rectangles marked with permanent marker. Swimming activity was monitored by counting the number of crossed lines in a 30-s observation period, every min for a total of 5 observation bins, over 5 min, according to Ponzoni et al. [36]. A webcam positioned in front of the observation tank video recorded the entire experiment for a later video-aided analysis. Three trained observers blind to treatment evaluated each video.

### 2.4. Statistical Analysis

#### 2.4.1. Mouse

In sensorimotor response experiments, data were expressed in arbitrary units (visual objects response, acoustic response, overall tactile reflex) and percentage of baseline (visual placing response). Data from spontaneous locomotion studies are expressed in absolute values for the total distance travelled (m). Startle amplitude was calculated as absolute values in decibel (dB). The amount of PPI was calculated as a percentage score for each prepulse + pulse trial type: % PPI = 100 − (([startle response for prepulse + pulse trial]/[startle response for pulse-alone trial]) × 100). Startle magnitude was calculated as the average response to all of the pulse-alone trials. All data are shown as mean ± SEM of 4 independent experimental replicates. Statistical analysis of the effects of each compound at different concentrations over time (Figure 2 and Figure 3, panels A–D) and those of the maximal effects (Figure 2 and Figure 3, panels E and F) was performed by two-way ANOVA followed by the Bonferroni post hoc test for multiple comparisons (for parametric data) and by Dunnett’s test for non-parametric data (score). Statistical analysis of the effects of DOB and PMA on startle amplitude and PPI (Figure 4) was performed with one-way ANOVA followed by the Bonferroni post hoc test for multiple comparisons. Also statistical analysis of the effect of DOB and PMA on spontaneous locomotion (Figure 5, panels A and B) was performed by two-way ANOVA followed by the Bonferroni post hoc test for multiple comparisons (for parametric data). Statistical analysis was performed using the program Prism software (GraphPad Prism, USA). * *p*  <  0.05, ** *p* < 0.01, *** *p*  <  0.001 versus vehicle. # *p* < 0.05, ## *p* < 0.01, ### *p* < 0.001 versus DOB.

#### 2.4.2. Zebrafish

Three observers who independently, through the videos, processed the scored behavior objectively and reliably quantified trance-like behavior. Apart from occasional exceptions, the three independent observers measured the same event. The results were statistically analyzed using the appropriate statistical test. To assess the differences among groups for swimming activity, a two-way analysis of variance (ANOVA) followed by Bonferroni’s post hoc tests were used. Hallucinatory behavior, evaluated as mean score of six observation bins, was analyzed by one-way non-parametric ANOVA (Kruskal–Wallis test) followed by Dunn’s post hoc test. All the statistical analyses were performed using Prism 6 software (Prism version 6, GraphPad Inc, La Jolla, CA, USA). * *p* < 0.05, ** *p* < 0.01, *** *p* < 0.001 **** *p* < 0.0001 versus vehicle.

## 3. Results

### 3.1. Evaluation of the Visual Object Response

Visual object response did not change in vehicle-treated mice in 5 h of observation (Figure 2, panels A and C), and the effect was similar to that observed in naïve untreated animals (data not shown). Systemic administration of DOB and PMA (0.01–30 mg/kg, i.p.) significantly (*p* < 0.0001) and dose-dependently reduced the visual object response in mice. DOB (significant effect of treatment (F_(5336)_ = 171.3), time (F_(7336)_ = 204.2), and time x treatment interaction (F_(35,336)_ = 12.11)) and PMA (significant effect of treatment (F_(5336)_ = 276.1), time (F_(7336)_ = 563.4), and time x treatment interaction (F_(35,336)_ = 23.0)) produced an impairment, especially at the highest dose of 30 mg/kg i.p., which persisted up to 5 h (Figure 2, panels A and C). In particular, DOB (Figure 2, panel A) significantly (*p* < 0.0001) inhibited, in a prolonged manner, the visual object response in mice at doses of 0.1–30 mg/kg, while PMA (Figure 2, panel C) significantly (*p* < 0.0001) inhibited, in a transient (1 mg/kg) or prolonged (10–30 mg/kg) manner, the visual object response. For both compounds, the lowest dose tested (0.01 mg/kg) did not significantly affect the visual object response. Comparison of the maximal effect between DOB and PMA (Figure 2, panel E) revealed a significant effect of treatment (F_(23,191)_ = 15.63, *p* < 0.0001). The effect of DOB was stronger than that induced by PMA at 10 and 30 mg/kg (*p* < 0.05) but not at 0.01–1 mg/kg.

### 3.2. Evaluation of the Visual Placing Response

Visual placing response did not change in vehicle-treated mice in 5 h of observation (Figure 2, panels B and D) and the effect was similar to that observed in naïve untreated animals (data not shown). Systemic administration of DOB and PMA (0.01–30 mg/kg, i.p.) significantly (*p* < 0.0001) and dose-dependently reduced the visual placing response in mice. DOB (significant effect of treatment (F_(5336)_ = 171.3), time (F_(7336)_ = 204.2), and time x treatment interaction (F_(35,336)_ = 12.11)) and PMA (significant effect of treatment (F_(5336)_ = 276.1), time (F_(7336)_ = 563.4), and time x treatment interaction (F_(35,336)_ = 23.0)) produced an impairment, especially at the higher doses of 10 and 30 mg/kg i.p., which persisted up to 5 h (Figure 2, panels B and D). In particular, DOB (Figure 2, panel B) significantly (*p* < 0.0001) inhibited, in a transient (0.01 mg/kg) or prolonged manner (0.1–30 mg/kg), the visual object response in mice, while PMA (Figure 2, panel D) significantly (*p* < 0.0001) inhibited, in a transient (1 mg/kg) or prolonged (10–30 mg/kg) manner, the visual object response. The visual object response was not affected by DOB at the lowest dose tested (0.01 mg/kg) and for PMA at 0.01 and 0.1 mg/kg. Comparison of the maximal effect between DOB and PMA (Figure 2, panel F) revealed a significant effect of treatment (F_(23,191)_ = 15.63, *p* < 0.0001). The effect of DOB was stronger than those induced by PMA only at 30 mg/kg (*p* < 0.05) but not at 0.01–10 mg/kg.

### 3.3. Evaluation of the Acoustic Response

Acoustic response did not change in vehicle-treated mice in 5 h of observation (Figure 3, panels A and C) and the effect was similar to that observed in naïve untreated animals (data not shown). Systemic administration of DOB (0.01–30 mg/kg, i.p.) significantly (*p* < 0.0001) and dose-dependently reduced the visual object response in mice (significant effect of treatment (F_(5336)_ = 171.3), time (F_(7336)_ = 204.2), and time x treatment interaction (F_(35,336)_ = 12.11)). DOB caused a rapid impairment of acoustic response, especially at the highest dose of 30 mg/kg i.p., which persisted up to 5 h (Figure 3, panel A). In particular, DOB significantly (*p* < 0.0001) inhibited, in a prolonged manner, the acoustic response in mice at doses of 1–30 mg/kg (Figure 3, panel A), while PMA significantly (*p* < 0.0001) inhibited, in a transient manner, the acoustic response in mice only at the highest dose (30 mg/kg; Figure 3, panel C) (significant effect of treatment (F_(5336)_ = 171.3), time (F_(7336)_ = 204.2), and time x treatment interaction (F_(35,336)_ = 12.11)). Comparison of the maximal effect between DOB and PMA (Figure 3, panel E) revealed a significant effect of treatment (F_(23,191)_ = 15.63, *p* < 0.0001). The effect of DOB was stronger than those induced by PMA at 10 and 30 mg/kg (*p* < 0.05) but not at 0.01–1 mg/kg.

### 3.4. Evaluation of the Overall Tactile Response

Overall tactile response did not change in vehicle-treated mice in 5 h of observation (Figure 3, panels B and D) and the effect was similar to that observed in naïve untreated animals (data not shown). Systemic administration of DOB (0.01–30 mg/kg, i.p.) significantly (*p* < 0.0001) reduced the visual object response in mice (significant effect of treatment (F_(5336)_ = 171.3), time (F_(7336)_ = 204.2), and time x treatment interaction (F_(35,336)_ = 12.11)). DOB caused a mild and transient impairment of tactile response at the higher doses of 10 and 30 mg/kg i.p., which persisted up to 4 h (Figure 3, panel B). PMA also significantly (*p* < 0.0001) affected overall tactile response in mice (significant effect of treatment (F_(5336)_ = 171.3), time (F_(7336)_ = 204.2), and time x treatment interaction (F_(35,336)_ = 12.11)). PMA inhibited, in a transient (1 mg/kg) or prolonged manner (10 mg/kg), the tactile response in mice, and biphasically affected the tactile response at the highest dose (30 mg/kg). In fact, PMA 30 mg/kg initially inhibited (up to 30 min; *p* < 0.001) and subsequently, facilitated (at 120 min; *p* < 0.05) the tactile response in mice (Figure 3, panel D). Comparison of the maximal inhibitory effect between DOB and PMA (Figure 3, panel F) reveals that there are no differences between DOB and PMA treatments (F_(23,191)_ = 15.63, *p* < 0.0001).

### 3.5. Startle and Prepulse Inhibition (PPI)

Vehicle injection did not change startle and PPI response in mice (Figure 4) and the effect was similar in naïve untreated animals (data not shown). Administration of DOB (0.01–10 mg/kg, i.p.) reduced startle amplitude in mice only at 10 mg/kg at 120 min (Figure 4, panel A). DOB at 10 mg/kg inhibited the PPI in mice after 15 min from drug administration (Figure 4, panel B). ANOVA analysis detected a significant reduction in prepulse intensity with 68 dB (~23%) (F_(3141)_ = 4.411, *p* = 0.0053), 75 dB (~27%) (F_(4970)_ = 6.397, *p* = 0.0005), and 85 dB (~27%) (F_(4970)_ = 6.397, *p* = 0.0005). Inhibition was also detected after 120 min from DOB administration (Figure 3, panel C). ANOVA analysis detected a significant reduction in prepulse intensity with 75 dB (~27%) (F_(4970)_ = 6.397, *p* = 0.0005) and 85 dB (~27%) (F_(4970)_ = 6.397, *p* = 0.0005). 

Administration of PMA (0.01–10 mg/kg, i.p.) did not affect startle amplitude in mice (Figure 4, panel D). PMA at 10 mg/kg inhibited the PPI in mice after 15 min from drug administration (Figure 4, panel E). ANOVA analysis detected a significant reduction in prepulse intensity with 68 dB (~23%) (F_(3141)_ = 4.411, *p* = 0.0053), 75 dB (~27%) (F_(4970)_ = 6.397, *p* = 0.0005), and 85 dB (~27%) (F_(4970)_ = 6.397, *p* = 0.0005). Inhibition was also detected after 120 min from PMA administration (Figure 3, panel F). ANOVA analysis detected a significant reduction in prepulse intensity only with 68 dB (~27%) (F_(4970)_ = 6.397, *p* = 0.0005). The lower doses of DOB and PMA (0.01–1 mg/kg) were ineffective on PPI response in mice.

### 3.6. Evaluation of the Spontaneous Locomotor Activity

With respect to vehicle-treated animals, DOB at 10 and 30 mg/kg reduced the total distance travelled in mice (Figure 5, panel A) (significant effect of treatment (F_(4720)_ = 6.6773, *p* < 0.0001), time (F_(15,720)_ = 84.28, *p* < 0.0001), and time x treatment interaction (F_(60,720)_ = 1.418, *p* = 0.0236)). The inhibitory effect was evident after 30 min and lasted up to 90 min.

Conversely, PMA biphasically affected the spontaneous locomotion in mice at highest doses, 10 and 30 mg/kg (Figure 5, panel B). In fact, PMA at 10 and 30 mg/kg initially inhibited (up to 30 min) and subsequently, facilitated the spontaneous locomotion in mice (significant effect of treatment (F_(4720)_ = 6.6773, *p* < 0.0001), time (F_(15,720)_ = 84.28, *p* < 0.0001), and time x treatment interaction (F_(60,720)_ = 1.418, *p* = 0.0236)). In particular, the facilitation induced by PMA at 30 mg/kg appeared earlier (effect significant at 75 min) than that induced by the dose of 10 mg/kg (effect significant at 90 min) and it was of longer duration (effect significant up to 135 min) with respect to those caused by the dose of 10 mg/kg (effect significant up to 105 min).

### 3.7. Effect of DOB and PMA in Zebrafish

Treatment with DOB and PMA modified normal swimming activity in zebrafish compared to the saline group in terms of appearance of trance-like behavior (Figure 6). The Kruskal–Wallis test revealed a significant difference between groups (DOB: H = 33.31, *p* < 0.0001; PMA: H = 45.69, *p* < 0.0001). Compared to the saline group, which showed no alterations, the fish treated with DOB and PMA (Figure 6) showed a significant reduction in the score. In particular, those exposed to 2 mg/kg of DOB showed slowed swimming, while PMA treatment progressively led to a trance-like behavior until a dose of 0.1 mg/kg. Increasing doses of PMA progressively returned to basal value, leading to a biphasic effect.

A significant decrease in the number of crossed lines after DOB and PMA treatment (Figure 7) was found. Two-way ANOVA showed differences among groups: DOB: treatment factor (F_(5360)_ = 35.14, *p* < 0.0001), time factor (F_(5360)_ = 0.43, *p* = 0.83), treatment x time _interaction_ (F_(25,360)_ = 0.75, *p* = 0.80); PMA: treatment factor (F_(5480)_ = 40.77, *p* < 0.0001), time factor (F_(5480)_ = 1.28, *p* = 0.27), treatment x time interaction (F_(35,480)_ = 1.85, *p* = 0.003). Post hoc analysis revealed a decreased number of crossed lines after treatment with DOB at all tested doses (0.075–2 mg/kg), and with PMA until the dose of 0.25 mg/kg. The dose of 0.5 mg/kg of DOB was the most effective.

## 4. Discussion

This in vivo study demonstrates that DOB and PMA impaired sensorimotor, motor, and sensory gating (prepulse inhibition) responses in mice, and promoted hallucinatory states in the zebrafish model by reducing the hallucinatory score and swimming activity.

DOB and PMA affected, in a dose-dependent manner, the visual, acoustic, and tactile sensory responses in mice, highlighting their psychoactive actions as found in humans [37,38].

Although DOB and PMA are structurally closely related (Figure 1, Panels A and B), DOB is more powerful and effective than PMA in inhibiting visual (both positioning and object; Figure 2) and acoustic responses (Figure 3, Panels A–C and E). Therefore, we may assume that their mechanisms of action are different. In fact, DOB is an agonist on 5-HT2 receptors, and in vitro, it was shown that DOB has the highest affinity for 5-HT2A (Ki = 0.6 ± 0.1 nM) compared to the other receptor subtypes (5-HT2B: Ki = 26.9 ± 4.6 nM and 5-HT2C Ki:1.3 ± 0.2 nM) [39]. In addition, the affinity of DOB for 5-HT2A and 5-HT2C is comparable to that of LSD (5-HT2A (Ki = 2.9 nM); 5-HT2C (Ki = 23 nM); [40] and their affinity is correlated with the amount of drug that induces hallucinogenic effects in humans [41]. These findings could explain the possible psychedelic/hallucinogenic effects induced by DOB, reflected on the long-lasting alterations of visual and acoustic responses, as reported with other psychedelic drugs such as 25I-NBOMe [42] and LSD [43]. In addition, the bromine existent at the paraposition on DOB chemical structure (Figure 1, panel A) could be responsible for reducing the number of metabolites, increasing its lipophilicity, and thus, its ability to cross the blood–brain barrier, and consequently, generating more powerful behavioral and psychotropic effects, as compared to PMA. [44].

Contrarily to DOB, the effects of PMA on brain neurotransmission are more similar to those of MDMA. In fact, PMA promotes the release of serotonin from storage sites [45] and has a potent ability to inhibit monoamine oxidase (MAO-A) [46], thus, facilitating the release of 5-HT in different areas of the brain. In fact, it has been demonstrated in microdialysis studies that PMA (5 and 10 mg/kg) enhances the release of 5-HT in rat striatum, nucleus accumbens, and frontal cortex [47,48]. However, a direct stimulation of PMA on 5-HT receptors was only hypothesized [49], since its structural analog PMMA binds to 5-HT receptors [48].

To a lesser extent, PMA is also a releaser and uptake inhibitor of DA and NE [50,51].

Serotonin-2A (5-HT2A) receptors have been involved in the sensory modulation and the induction of hallucination in humans [52] and rodents [53].

Our results showed clearly how DOB and PMA reduced, in a dose-dependent manner, the visual responses of mice to the object and the placing (Figure 2). DOB demonstrated a higher potency than PMA in both tests (Figure 2A–D). The alteration of visual reflexes by stimulant/psychedelic drugs shows conflicting results in studies of firing rate of individual V1 neuron responses to visual stimuli, where it has been demonstrated that firing rate could be facilitated, suppressed or both, after the administration of 5-HT2A agonists [53,54,55,56]. Yet, in a recent study using wide field and two-photon calcium imaging and single-unit electrophysiology in awake mice, it has been established that administration of 2,5-dimethoxy-4-iodoamphetamine (DOI) at the dose of 10 mg/kg leads to a net reduction in visual response amplitude and surround suppression in the primary visual cortex, as well as disrupted temporal dynamics [57]. These data confirm the inhibitory effects of DOB and PMA on visual reflexes, and prove that both drugs could impair cortical circuits, possibly through the activation of 5HT2A-R, resulting in perceptual deficits [57].

DOB and PMA showed a dose-dependent inhibition also of the acoustic response in mice (Figure 3, panels A and C). The comparison between the maximal effects of the two molecules shows once again the greater potency of DOB in comparison to PMA (Figure 3, Panel E), especially at higher doses (10–30 mg/kg, i.p.). Results are in accordance with previous studies conducted with other hallucinogenic substances as DOI [58] and 25I-NBOMe [42] on rats, or MDMA [29] in mice.

The role of serotonin in the modulation of auditory responses in various species has been established [59]. In rodents, serotonergic fibers are abundant in cochlear nucleus (CN) and the inferior colliculus (IC), which reveal their importance in auditory processing [60,61]. In fact, the administration of serotonin receptor agonists induces changes in response threshold to sounds [62,63]. In particular, DOI induces an inhibitory effect on the spike count on IC neurons [64]. Given the abovementioned mechanisms of action of serotonin receptor agonists, DOB and PMA could also impair the serotoninergic signals involved in the auditory responses.

DOB and PMA do not seem to be very effective in reducing overall tactile sensory stimuli: pinnae, vibrissae, and corneal (Figure 3, panels B and D), as recorded in visual and acoustic responses. Our data show that there is no significant difference between the maximal effects induced by DOB and PMA on tactile responses in mice (Figure 3, panel F). Instead, it is interesting to note how the two compounds modulated the tactile response differently: DOB inhibits mice tactile responses only at the highest dose tested (30 mg/kg; Figure 2, panel B). The results are in line with previous studies with 25I-NBOMe [42] in rats, and with MDMA in both rats and mice [29]. The inhibitory effect of DOB on tactile response could be related to the stimulation of 5-HT2A receptors in the cortical areas of the brain [65]. Inversely to DOB, PMA was effective at lower doses (1 mg/kg; Figure 2, panel D). Moreover, PMA shows a biphasic effect at the highest dose tested, which cannot be directly attributed to its hallucinogenic action. Yet, the increase in tactile response, as for MPA [27], could be related to its facilitation of DA and NE [51] release.

To better understand the sensory alterations induced by both DOB and PMA, animals were tested on PPI at 0.01–10 mg/kg. Systemic administration of DOB reduced upon 120 min the acoustic startle reflex in mice, but only at the dose of 10 mg/kg (Figure 4, panel A). Even if the mouse model does not seem promising for the PPI test like rats, our study is in accordance with previous studies demonstrating the disruption of PPI in rodents after the administration of different hallucinogenic compounds as 25I-NBOMe [42], LSD [12], and DOI [66]. Moreover, it has been demonstrated that DOI produces a disruption of PPI when infused into the ventral pallidum (VP). The infusion of M100907 (5-HT2A antagonist) into the VP blocked the PPI disruption induced by DOI. These findings reveal the role of 5-HT2A on the modulation of sensorimotor gating in mice [67].

Differently to DOB, PMA caused a moderate effect on the startle reflex in mice, however, just at the highest dose tested (10 mg/kg; Figure 4, panels D and F). In our previous study, we demonstrated that MDMA at 10 mg/kg did not alter PPI in mice, however, a significant disruption was measured at the dose of 20 mg/kg. The inhibitory effect of PMA on the PPI test depends on the dose injected [29].

Obtained data showed opposite results on spontaneous locomotion: DOB induced significant inhibition of the distance travelled at higher doses (10 and 30 mg/kg; Figure 5, panel A), while PMA induced a biphasic effect at the two highest doses tested (10–30 mg/kg; Figure 5, panel B).

Our results on DOB are not in accordance with previous studies, as it has been demonstrated that phenylalkylamine hallucinogens as DOI and DOM induces a bell-shaped dose–response [68,69], where the dose of 1 mg/kg facilitates locomotion, and the dose of 10 mg/kg reduces. This difference in results could be related to the method used and the animal strains. In addition, it has been demonstrated that the inhibitory effect induced by DOI has been blocked by 5-HT2C agonist (SER-082), which seems to reveal the role of these receptors on locomotor activity of mice [68].

Past studies confirmed that unlike MDMA [70], PMA does not show a strong enough action on the locomotor system, which could be related to the fact that PMA is a weak DA releaser [70,71]. In fact, studies in mice confirm stimulating effects on locomotor activity of PMA only at the higher dose of 30 mg/kg [72,73], but not at lower doses [70]. Studies in rats with PMA showed a dose-dependent trend of stimulation [74]. In line with our results, PMMA induced a biphasic effect on rats: at doses of 5 and 20 mg/kg, an initial ataxia followed by locomotor stimulation [75].

Regarding zebrafish, it is necessary to rely on recent studies conducted by Ponzoni and his colleagues, according to which the effects of DOB and PMA in Conditioned Place Preference (CPP) tests are predictive of the potential abuse of these substances, and its “trance-like” behavior typical of hallucinogenic substances [76]. Notably, the rewarding and hallucinatory effect was blocked by ritanserin, a serotonin 5-HT2 receptor subtype antagonist, suggesting the involvement of the serotonergic system [30].

The trance-like behavior evoked by PMA showed a biphasic trend. In zebrafish, this pattern has also been observed for other effects induced by DOB and PMA, like pro-social, anxiolytic, and rewarding [30,76]. The observed inverted U-shaped curve is also in line with the findings obtained in other animal models, and in some human studies [77]. Our results fit well with the obtained results in PPI, at which mice had already demonstrated sensory alterations after DOB and PMA treatment. PPI in rodents was altered after the administration of some hallucinogenic compounds.

The decreased number of crossed lines indicates a decrease in swimming activity. At least for PMA, which exhibited the highest potency, this effect was probably due to the appearance of trance-like behavior. The reduced swimming activity is in line with the findings obtained in locomotor activity in mice, suggesting the validity of the zebrafish model. Collectively, the observed effects of DOB and PMA on fish, strikingly resemble those in humans, further supporting the translational value of zebrafish model for psychedelic drug research.

## 5. Conclusions

These data suggest that the observed effects of DOB and PMA on both mouse and fish models are surprisingly similar to those in humans. This study appears to be a further confirmation of the validity of the zebrafish model for preclinical research of psychedelic drugs.

## Figures and Tables

**Figure 1 brainsci-10-00586-f001:**
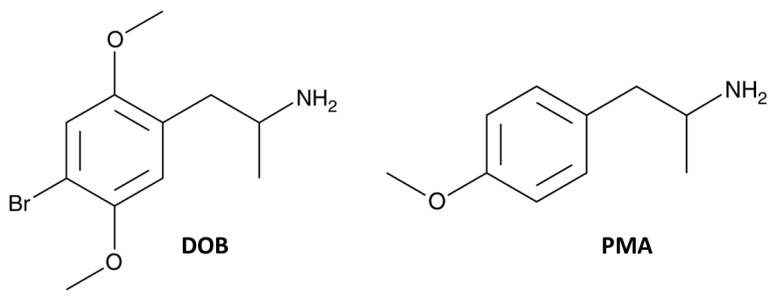
Chemical structures of DOB (2, 5-dimetoxy-4-bromo-amphetamine) and PMA (paramethoxyamphetamine).

**Figure 2 brainsci-10-00586-f002:**
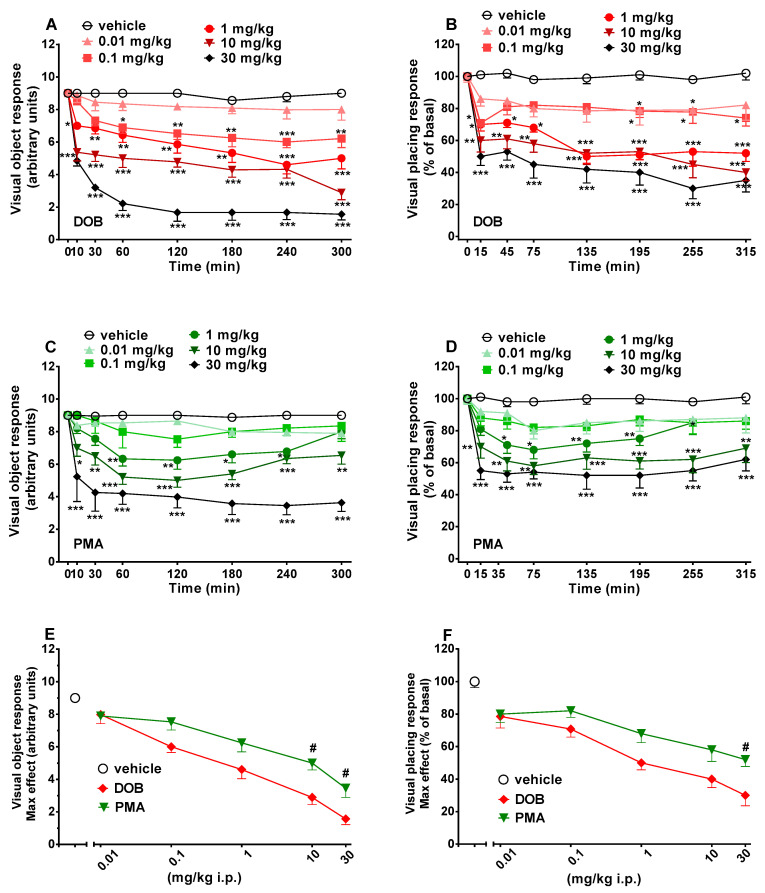
Effect of DOB (0.01–30 mg/kg i.p.; panels **A** and **B**) and PMA (0.01–30 mg/kg i.p.; panels **C** and **D**) on the visual object (**left**) and placing response (**right**) tests in the mice and comparison of the maximum effect observed in 5 h on visual object (panel **E**) and visual placing (panel **F**) tests. Data are expressed as mean ± SEM (*n* = 8/group). Statistical analysis was performed by two-way ANOVA followed by Bonferroni’s test (panels **B**,**D**,**E**), Dunnett’s test (panels **A**,**C**,**F**) for multiple comparisons for the dose–response curve of each compound at different time points (panels **A**–**D**). The comparison of maximum effect observed in 5 h was also presented (panels **E** and **F**). * *p* < 0.05, ** *p* < 0.01, *** *p* < 0.001 versus vehicle; # *p* < 0.05 versus DOB.

**Figure 3 brainsci-10-00586-f003:**
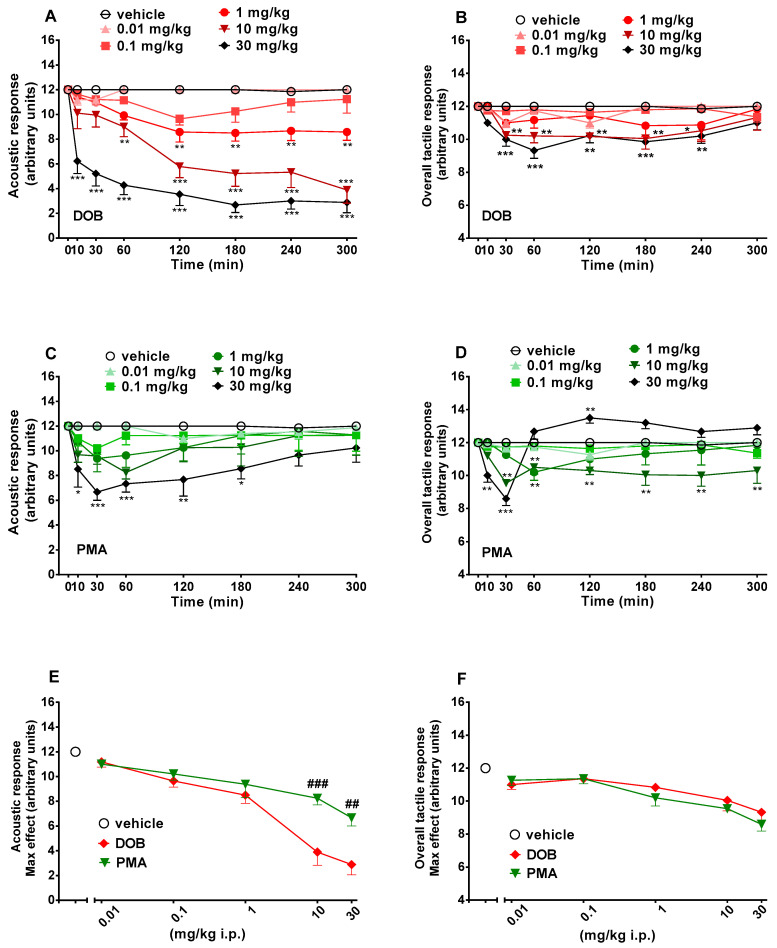
Effect of DOB (0.01–30 mg/kg i.p.; panels **A** and **B**) and PMA (0.01–30 mg/kg i.p.; panels **C** and **D**) on the acoustic (**left**) and the overall tactile (**right**) response in the mice and comparison of the maximum effect observed in 5 h on visual object (panel **E**) and visual placing (panel **F**) tests. Data are expressed as mean ± SEM (*n* = 8/group). Statistical analysis was performed by two-way ANOVA followed by Dunnett’s test for multiple comparisons for the dose–response curve of each compound at different time points (panels **A**–**D**) and for the statistical analysis of the maximum effect observed in 5 h (panels **E** and **F**). * *p* < 0.05, ** *p* < 0.01, *** *p* < 0.001 versus vehicle; ## *p* < 0.01, ### *p* < 0.001 versus DOB.

**Figure 4 brainsci-10-00586-f004:**
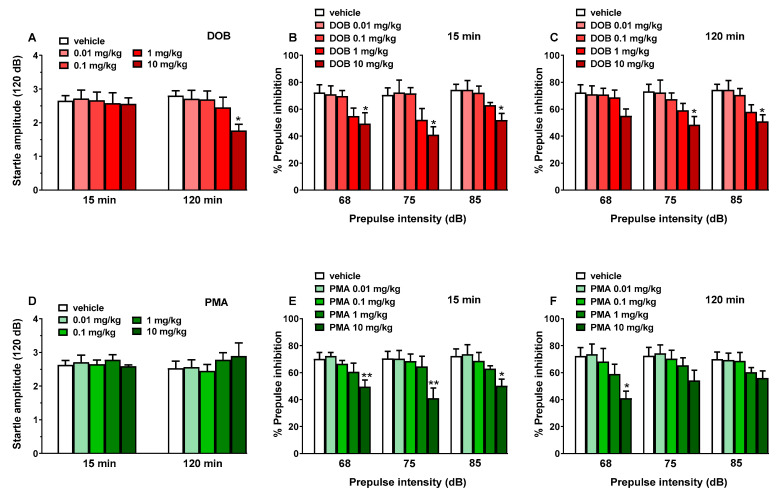
Effect of DOB (0.01–10 mg/kg i.p.; panels **A**–**C**) and PMA (0.01–10 mg/kg i.p.; panels **D**–**F**) on startle amplitude (panels **A** and **D**) and prepulse inhibition (PPI) in mice (panels **A**,**B**,**E**,**F**). Effects on PPI are shown for the three prepulse intensities (68, 75, and 85 dB), 15 min (panels **B** and **E**), and 120 min (panels **C** and **F**) after treatment. Startle amplitude was expressed in absolute values (in dB) and prepulse inhibition (PPI) was expressed as the percentage decrease in the amplitude of the startle reactivity caused by presentation of the prepulse (% PPI; see Material and Methods) and values represent mean ± SEM of 10 animals for each treatment. The statistical analysis was performed with a one-way ANOVA followed by Bonferroni’s test for multiple comparisons. * *p* < 0.05 and ** *p* < 0.01 versus vehicle.

**Figure 5 brainsci-10-00586-f005:**
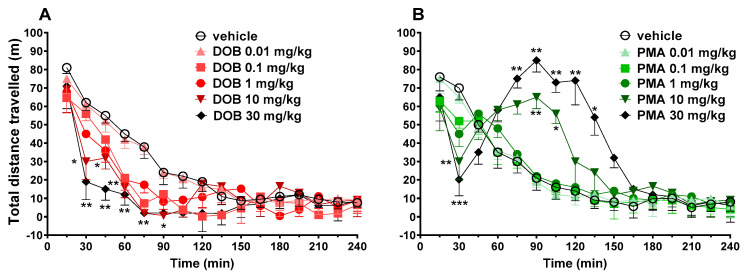
Effect of DOB (0.01–30 mg/kg i.p.; panel **A**) and PMA (0.01–30 mg/kg i.p.; panel **B**) on the total distance travelled of mice in 4-h observation. Data are expressed as meters travelled and represent the mean ± SEM of 8 determinations for each treatment. Statistical analysis was performed by two-way ANOVA followed by Bonferroni’s test for multiple comparisons for the dose–response curve of each compound at different times. * *p* < 0.05, ** *p* < 0.01 and *** *p* < 0.001 versus vehicle.

**Figure 6 brainsci-10-00586-f006:**
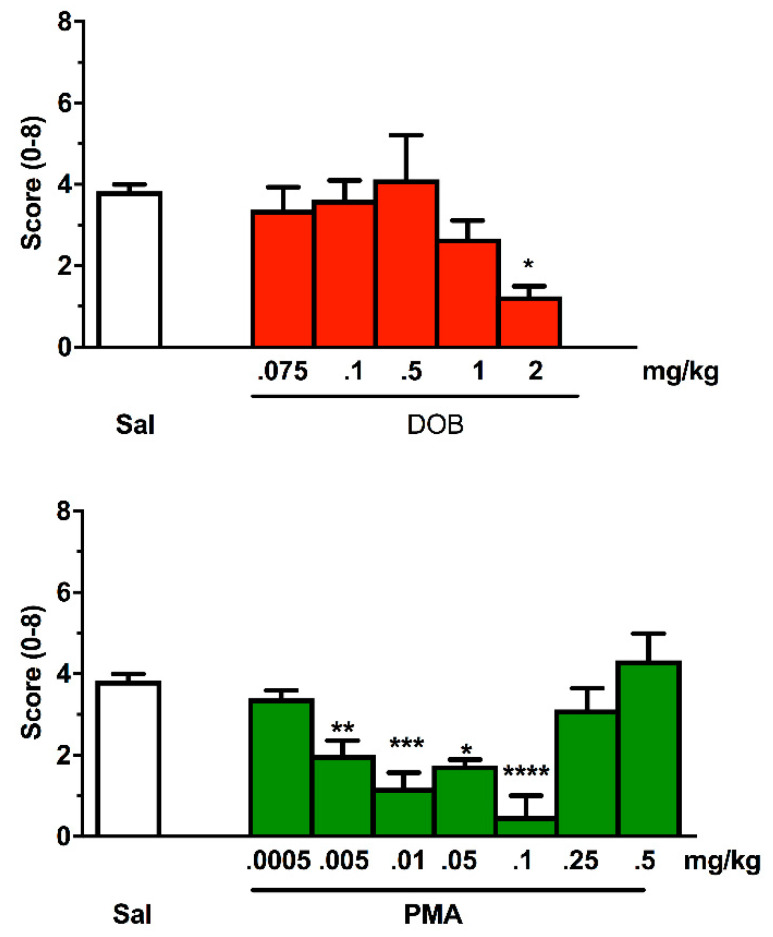
Effect of increasing doses of DOB and PMA on the appearance of trance-like effect evaluated for 5 min after treatment. Each compound was given i.m. immediately before the test. *n* = 10 for each group. Data were expressed as score (±SEM). * *p* <  0.05, ** *p* < 0.01, *** *p* < 0.001, **** *p* < 0.0001, compared with the corresponding saline group (Sal) (Kruskal–Wallis, Dunn’s test).

**Figure 7 brainsci-10-00586-f007:**
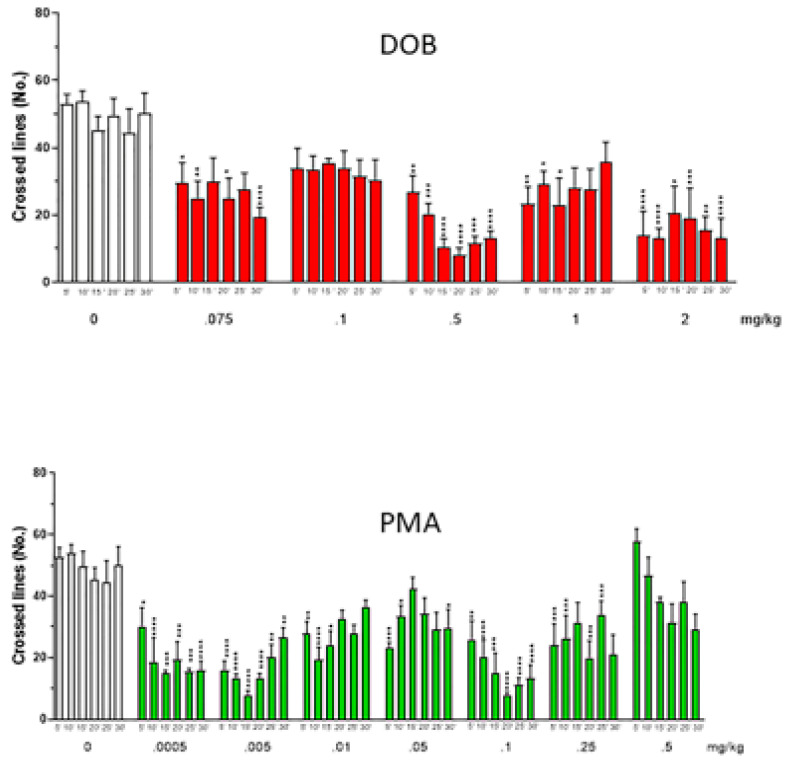
Effect of increasing doses of DOB and PMA on the number of crossed lines evaluated every min, for 5 min after treatment. Each compound was given i.m. immediately before the test. N = 10 for each group. Data were expressed as mean (±SEM). * *p* <  0.05, ** *p* < 0.01, *** *p* < 0.001, **** *p* < 0.0001 compared with the corresponding saline group (Sal).

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
