# Peer review of "Acute DOB and PMA Administration Impairs Motor and Sensorimotor Responses in Mice and Causes Hallucinogenic Effects in Adult Zebrafish"

_brainsci, 2020, doi:10.3390/brainsci10090586_

Round 1

Reviewer 1 Report

This is a nice study on investigate the effect of the administration of DOB and PMA. Finding from the mice study is encouraging. This study can be a good addition to the current psychopharmacological research. Zebrafish study plan can add to the suitability of the tiny animal for drug discovery and in vivo screening of hallucinogens. However, I am having some concerns with the study which I stated bellow.

  1. Author need to be carefully with typos. In the 1st para of introduction section line# “such as” is written as “such us”. Please avoid such error in the 1st para of the manuscript.
  2. It was stated zebrafish tank were maintained under standard conditions at 28 (± 2) °C. Zebrafish is very sensitive to surrounding temperature. 26-30 °C variation is very high for that tiny animal. You should try to maintain the temperature at  28 °C. Otherwise temperature variation will stressed the fish.
  3. Did the authors tried to standardized the dose effect on zebrafish? The ref #30 is was on mice and I failed to find the mentioned dose range. If there is no previous report author should fast have standardized the dose range.
  4. There exist lots of out of context referencing. One example is the authors mentioned “Zebrafish body weight was measured as previously described [26].” Ref#26 has nothing to do with Zebrafish body weight. Actual ref could be Novak et al.
  5. Where is Figure 7. What is figure 6 a and b. Authors need to be serious about their work presentation.
  6. To come to a conclusion on hallucinogenic effect of zebrafish author need to perform novel tank test, social behavior in the shoaling test.

Finally, authors have performed the mouse study with systematic approach and describe that nicely. However, same is missing in the zebrafish study.

Author Response

In reply to Reviewer #1

- We thank the Reviewer for his/her positive evaluation of our manuscript and for helpful suggestions to improve the article. Our reply to each specific comment is in italic, blue and indicated by R:. We also reported line number of the changes and we highlighted in yellow the new parts in the MS file (brainsci-893124 REW).

Q1: Author need to be carefully with typos. In the 1st para of introduction section line# “such as” is written as “such us”. Please avoid such error in the 1st para of the manuscript.

R1: We thank the Reviewer; we have corrected the error.

Q2: It was stated zebrafish tank were maintained under standard conditions at 28 (± 2) °C. Zebrafish is very sensitive to surrounding temperature. 26-30 °C variation is very high for that tiny animal. You should try to maintain the temperature at 28 °C. Otherwise temperature variation will stressed the fish.

R2: In the “Guidance of the housing and care of Zebrafish Danio rerio “ by Barney Reed & Maggy Jennings (Research Animals Department,Science Group,RSPCA) a water temperature of 28.5° is widely cited as the optimum temperature for zebrafish. However, temperatures from 24° to 30° are also reported.

Q3: Did the authors tried to standardized the dose effect on zebrafish? The ref #30 is was on mice and I failed to find the mentioned dose range. If there is no previous report author should fast have standardized the dose range.

R3: In line 133 the reference, in which the dose effect of the drugs is reported, the number is 31 and not 30

Q4: There exist lots of out of context referencing. One example is the authors mentioned “Zebrafish body weight was measured as previously described [26].” Ref#26 has nothing to do with Zebrafish body weight. Actual ref could be Novak et al.

R4: Line 187 reference is 27 and not 26. We correctly inserted the numeration of references but the automatic formatting of references starts considering the section item (references) as n ° 1. So all the references are down one place. Ref 1 becomes ref 2 by mistake of the newspaper formatting.

Q5: Where is Figure 7. What is figure 6 a and b. Authors need to be serious about their work presentation.

R5: We apologize to the Reviewer and Figure 7 is provided. “a” and “b” have been eliminated

Q6: To come to a conclusion on hallucinogenic effect of zebrafish author need to perform novel tank test, social behavior in the shoaling test.

R6: Social behaviour and shoaling have been already published in our previous work: PONZONI L, SALA M, BRAIDA D. Ritanserin-sensitive receptors modulate the prosocial and the anxiolytic effect of MDMA derivatives, DOB and PMA, in zebrafish. Behavioural Brain Research 2016. 314, 181

Q7: Finally, authors have performed the mouse study with systematic approach and describe that nicely. However, same is missing in the zebrafish study.

R7: We agree with the Reviewer that visual, acoustic, tactile responses and PPI should be do also in zebrafish. However, to reduce the number of animals we decided to do these three tests only in mice.

Reviewer 2 Report

In this study, the psychoactive substances 2,5-dimetoxy-4-bromo-amphetamine (DOB) and para-methoxyamphetamine (PMA) were administered systemically on zebrafish and male mice to investigated pro-psychedelic/hallucinogenic effects on motor/sensorimotor behaviors. It is recognized the interest and relevance of the subject. Manuscript is satisfactorily written. Several parts were highlighted as confusing in their message. Reading was relatively easy to follow. Revision by a native English speaker is recommended. In terms of experimental methodology and results, manuscript is fine executed, although some questions/details require clarification. Ethical statement is provided. Nevertheless, some concerns were highlighted that must be covered. Discussions and conclusions are relevant and supported by collected data. Statistical analysis is provided, and just a few minor details have to be revised. Figures and Tables are pertinent and detailed.  Major questions/concerns were highlighted through the manuscript document enclosed.

Author Response

In reply to Reviewer #2

- We thank the Reviewer for his/her positive evaluation of our manuscript and for helpful suggestions to improve the article. Our reply to each specific comment is in italic, blue and indicated by R:. We also reported line number of the changes and we highlighted in yellow the new parts in the MS file (brainsci-893124 REW).

As requested by the Reviewer, we have changed and corrected all the parts highlighted in yellow in the manuscript and we have responded to the comments.

Comment 1 (Line 82): While in rodents it is very difficult to evaluate hallucinations (given that ...) - edit the sentence as recommended and complete it, explaining why it is difficult.

We removed the sentence ("At difference of rodents where is very difficult to evaluate hallucinations") since evaluation of hallucinogenic effect of compound in mice is possible by measuring the Head Twitch test.

Comment 2 (Line 89): Confusing …

We have simplified the sentence

Comment 3 (Line 94): Why only males were used? - rationale should be indicated

We only used males as we did not investigate gender differences in this research. Gender difference studies will be completed shortly.

Comment 4 (Line 96): How the number os required mice were calculated? was 3Rs Principle taked into account? - statistics performed to define the number of animals to be used must be referred.

We fully agree with the reviewer. We apologize for the mistake we correct. The number of mice per cage is 5 animals. This number of animals per cage is defined by the housing rules of the Centralized Laboratory for Preclinical Research (LARP, University of Ferrara) where the animals are raised and maintained.

As required by Reviewer, we inserted (at line 143) the following sentence in the manuscript: “The number of animals used was defined according to the 3R rules. In particular, the determination of the number of animals to be used (sample size) and the calculation of adequate power in the statistical analysis of the data (power analysis) was determined using the simulation software G * Power 3.1.9.2”

Comment 5 (Line 104): Adult wild-type short fin zebrafish (Why?).

Adult wild-type short fin zebrafish (Danio Rerio), (strain usually used for the zebrafish sequencing project because it is cleaned up to remove embryonic lethal mutations from the background).

Comment 6 (Line 109-110): -This is a tolerable range, not zebrafish optmial pH environment which should be between pH 7.0 and 8.0 to promote good health of biofilters and stable water quality.

We agree with the Reviewer and we inserted the correct form with the following sentence: “Zebrafish are maintained with a pH ranging from 6.6 to 8.2”

-nitrites measurements are more relevant.

We agree with the Reviewer and we inserted the nitrites measurements “for nitrates (<0.02 ppm) and nitrites (0.05 e 0.15 mg/L)”

Comment 7 (Line 123):  add referece(s) confirming that at this ethanol and Tween 80 nominal concentrations, no toxicity is recorded on mouse models.

We agree with the Reviewer and we added the references Bilel et al 2020 and

De-Giorgio et al., 2020

Comment 8 (Line 226):  were the parametric assumptions (homogeneity of variances and normal distribution) met? - indicate

We agree with the Reviewer, and we apply the correct statistical analysis following the criteria of homogeneity of variances and normal distribution. Where data distribution was normal and variance comparable two-way ANOVAs (with repeated measures when appropriate), followed by Bonferroni's post hoc tests were used. Non-parametric data (score) were analyzed by two-way ANOVAs, followed by Dunnett’s test. We have also corrected the figure legends.

Comment 9 (Line 230):  P value is missing - indicate

We reported P values in figure legends. However, we also inserted P values as requested by Reviewer. *P < 0.05, **P < 0.01, ***P < 0.001 versus vehicle. #P<0.05, ##P<0.01, ###P<0.001 versus DOB

Comment 10 (Line 235):  confusing…

We agree with the Reviewer and we corrected the sentences. We deleted: "and, if no significance (p>0.05) was obtained, data were put together"

Comment 11 (Line 236):  were the parametric assumptions (homogeneity of variances and normal distribution) met? - indicate

Where data distribution was normal and variance comparable (swimming activity) two-way ANOVAs (with repeated measures when appropriate), followed by Tukey's or Bonferroni's post hoc tests were used. Non-parametric data (score) were analyzed by Kruskall-Wallis test.

Comment 12 (Line 337):  used "H" instead

We have corrected

Comment 13 (Line 403):  add reference(s)

We remove the word “partial” (we have not clear evidence of this pharmacodynamic aspect)….and have added the reference.

Comment 14 (Line 420):  confusing

We changed the sentence “since its structural analog, PMMA, binds to 5-HT receptors”.

Comment 15 (Line 433):  add reference(s)

We have added the reference.

Comment 16 (Line 485):  confusing

We have change the sentence: “Past studies confirmed that unlike MDMA [70], PMA does not show a strong enough action on the locomotor system, which could be related to the fact that PMA is a weak DA releaser [70-71]. In fact, studies in mice confirm stimulating effect on locomotor activity of PMA only at higher dose of 30 mg/kg [72-73] but not at lower doses [70]”.

Comment 17 (Line 498):  add reference(s)

We have added the reference: Braida D, Donzelli A, Martucci R, Ponzoni L, Pauletti A, Sala M. Neurohypophyseal hormones protect against pentylenetetrazole-induced seizures in zebrafish: role of oxytocin-like and V1a-like receptor. Peptides. 2012 Oct;37(2):327-33.

Round 2

Reviewer 1 Report

Majority of my concerns has been addressed except the first one.

Reviewer 2 Report

In this study, the psychoactive substances 2,5-dimetoxy-4-bromo-amphetamine (DOB) and para-methoxyamphetamine (PMA) were administered systemically on zebrafish and male mice to investigated pro-psychedelic/hallucinogenic effects on motor/sensorimotor behaviors. Authors took into consideration every recommendation previously pointed.